# Altered Ca^2+^ Homeostasis in Red Blood Cells of Polycythemia Vera Patients Following Disturbed Organelle Sorting during Terminal Erythropoiesis

**DOI:** 10.3390/cells11010049

**Published:** 2021-12-24

**Authors:** Ralfs Buks, Tracy Dagher, Maria Giustina Rotordam, David Monedero Alonso, Sylvie Cochet, Emilie-Fleur Gautier, Philippe Chafey, Bruno Cassinat, Jean-Jacques Kiladjian, Nadine Becker, Isabelle Plo, Stéphane Egée, Wassim El Nemer

**Affiliations:** 1BIGR, UMR_S1134, Inserm, Université de Paris, F-75015 Paris, France; ralfs.buks@abdn.ac.uk (R.B.); sylvie.cochet@inserm.fr (S.C.); 2Institut National de la Transfusion Sanguine, F-75015 Paris, France; 3Laboratoire d’Excellence GR-Ex, F-75015 Paris, France; tracydagher@gmail.com (T.D.); david.monedero-alonso@yale.edu (D.M.A.); ef.gautier.upd@gmail.com (E.-F.G.); bruno.cassinat@aphp.fr (B.C.); jean-jacques.kiladjian@aphp.fr (J.-J.K.); isabelle.plo@gustaveroussy.fr (I.P.); egee@sb-roscoff.fr (S.E.); 4U1287, Inserm, Université Paris-Saclay, Gustave Roussy, F-94800 Villejuif, France; 5Nanion Technologies GmbH, 80339 Munich, Germany; Maria.Rotordam@nanion.de (M.G.R.); Nadine.Becker@nanion.de (N.B.); 6Theoretical Medicine and Biosciences, Medical Faculty, Saarland University, Kirrbergerstr. 100, DE-66424 Homburg, Germany; 7Sorbonne Université, CNRS, UMR LBI2M, Station Biologique de Roscoff SBR, F-29680 Roscoff, France; 8Institut Imagine-INSERM U1163, Necker Hospital, Université de Paris, F-75015 Paris, France; 9Proteomics Platform 3P5, Université de Paris, Institut Cochin, INSERM, U1016, CNRS, UMR8104 Paris, France; philippe.chafey@inserm.fr; 10IRSL, U1131, INSERM, Université de Paris, F-75010 Paris, France; 11Hôpital Saint-Louis, Laboratoire de Biologie Cellulaire, AP-HP, F-75010 Paris, France; 12Centre d’Investigations Cliniques, Hôpital Saint-Louis, Université de Paris, F-75010 Paris, France; 13Etablissement Français du Sang PACA-Corse, F-13005Marseille, France; 14Aix Marseille Univ, EFS, CNRS, ADES, “Biologie des Groupes Sanguins”, F-13005 Marseille, France

**Keywords:** polycythemia vera, JAK2V617F, red blood cells, Ca^2+^, organelle sorting, enucleation, reticulocytes

## Abstract

Over 95% of Polycythemia Vera (PV) patients carry the V617F mutation in the tyrosine kinase Janus kinase 2 (JAK2), resulting in uncontrolled erythroid proliferation and a high risk of thrombosis. Using mass spectrometry, we analyzed the RBC membrane proteome and showed elevated levels of multiple Ca^2+^ binding proteins as well as endoplasmic-reticulum-residing proteins in PV RBC membranes compared with RBC membranes from healthy individuals. In this study, we investigated the impact of JAK2^V617F^ on (1) calcium homeostasis and RBC ion channel activity and (2) protein expression and sorting during terminal erythroid differentiation. Our data from automated patch-clamp show modified calcium homeostasis in PV RBCs and cell lines expressing JAK2^V617F^, with a functional impact on the activity of the Gárdos channel that could contribute to cellular dehydration. We show that JAK2^V617F^ could play a role in organelle retention during the enucleation step of erythroid differentiation, resulting in modified whole cell proteome in reticulocytes and RBCs in PV patients. Given the central role that calcium plays in the regulation of signaling pathways, our study opens new perspectives to exploring the relationship between JAK2^V617F^, calcium homeostasis, and cellular abnormalities in myeloproliferative neoplasms, including cellular interactions in the bloodstream in relation to thrombotic events.

## 1. Introduction

Polycythemia Vera (PV) and Essential Thrombocythemia (ET) are hematological disorders characterized by clonal expansion of abnormal hematopoietic stem cells leading to excessive proliferation of red blood cells and platelets, respectively. Over 95% of PV patients and 60% of ET patients display the V617F mutation in the tyrosine kinase Janus kinase 2 (JAK2) [1]. The JAK2^V617F^ mutation causes both quantitative and qualitative blood cell abnormalities [2]. Thrombotic complications constitute the main cause of morbidity and mortality, accounting for a 45% mortality rate in PV [3].

The production of mature RBCs from hematopoietic stem cells is a complex and tightly regulated process. A hallmark of mammalian RBCs is their lack of nucleus following DNA condensation and nucleus extrusion during terminal erythroid differentiation [4]. Our proteome analysis of PV RBC membranes showed abnormal presence or increased levels of endoplasmic reticulum (ER) residing proteins, such as ER chaperone 78-kDa glucose-regulated protein (GRP78) and Peroxiredoxin 4 (Prx4), the only ER localized peroxiredoxin [5]. The presence of organelle proteins in mature RBC membranes suggests affected organelle sorting during enucleation and/or affected reticulocyte maturation in a JAK2^V617F^ context. Organelle remnants in circulating reticulocytes and mature RBCs could affect the properties of blood, potentially contributing to thrombotic event formation.

Ca^2+^ ions are key players in a vast array of processes in circulating RBCs including thrombus formation [6]. Our proteome analysis of PV RBCs showed elevated levels of Ca^2+^ binding proteins, such as calnexin and calreticulin, suggesting modified calcium homeostasis in these cells [5]. Potentially altered Ca^2+^ homeostasis could affect RBC properties and plays a role in PV pathophysiology. Despite thrombosis being the main cause of mortality in PV patients, Ca^2+^ homeostasis in mature PV RBCs has not been explored.

In this study, we investigated calcium homeostasis in human and mouse JAK2^V617F^ RBCs by directly measuring intracellular Ca^2+^ levels and by assessing the activity of the Ca^2+^-activated Gárdos channel (KCNN4). Additionally, we investigated the role of JAK2^V617F^ in terminal erythroid differentiation, focusing on organelle sorting during enucleation. Finally, we performed a proteomics analysis to quantify and compare protein levels in mature RBCs and reticulocytes from PV and ET patients with JAK2^V617F^ mutation and from healthy individuals.

## 2. Materials and Methods

### 2.1. Human Blood Samples

The study obtained ethical approval from the Comité de Protection des Personnes Ile de France VII (PP 14-035). We used fresh peripheral blood samples from aspirin-treated or non-treated JAK2^V617F^ PV and ET patients, and healthy donors. The erythroid cells were selected by cellulose separation following density gradient separation, enriching the samples in reticulocytes, as previously described [7]. Control RBCs and RBCs from JAK2^V617F^ PV patients were cryopreserved and stored in the rare blood collection at the Centre National de Référence des Groupes Sanguins (Paris, France). Cryopreserved RBCs were thawed and shipped in Cell Stab (Bio-Rad, Hercules, CA, USA). Informed consent for research studies was obtained for all subjects in accordance with the Declaration of Helsinki.

### 2.2. Mice Samples

Animal experiments were approved by the Institut Gustave Roussy review board, protocol no. 2016-026 (APAFIS#7171-2016101015023392). A JAK2^V617F^ conditional knock-in (KI) mouse model was generated using conditional floxed JAK2 (JAK2^FLEX/+^) KI mice backcrossed over 15 times with C57Bl/6 mice [8]. To express the mutation, KI mice were crossed with VavCre transgenic (TG) mice [9]. VavCre TG mice targets in vivo Cre recombination in hematopoietic stem cells [10] and some endothelial cells [11]. Flushed bone marrow and spleen cell suspension from six JAK2^WT^ and six JAK2^V617F^ mice littermates was prepared. Peripheral blood from at least five mice in each condition was collected and used for analysis.

### 2.3. Flow Cytometry

The CaCl_2_ solution (Sigma-Aldrich, St. Louis, MO, USA) and RBC osmotic buffer (137 mM NaCl, 2.7 mM KCl, 2.05 mM MgSO_4_, glucose 5 mM (Sigma-Aldrich, St. Louis, MO, USA), 1.5 mM KH_2_PO_4_ (Merck Group, Darmstadt, Germany), 4.72 mM NaH_2_PO_4_ (Prolabo, Paris, France); pH = 7.4, 300 mOsm) were prepared in-house. The RBCs were washed three times with the osmotic buffer. Free intracellular Ca^2+^ was measured in an osmotic buffer using Fluo4-AM dye (Life Technologies, Carlsbad, CA, USA) in RBCs from healthy individuals (CT) and PV patients using BD FACSCanto™ II flow cytometry system (BD Biosciences, Franklin Lakes, NJ, USA). K_Ca_3.1 (Gárdos) levels were measured by anti-K_Ca_3.1-ATTO-488 (Alomone labs, Jerusalem, Israel) antibody in CT and PV RBCs, and BaF3 EpoR JAK2^WT^ and BaF3 EpoR JAK2^V617F^ cells.

Human and mice peripheral blood samples were washed three times with PBS (Thermo Fisher Scientific, Waltham, MA, USA). Samples were stained with thiazole orange in PBS with 0.5% BSA (Sigma-Aldrich, St. Louis, MO, USA) for 30 min at room temperature (RT). Mice samples were processed using BD FACSMelody™ cell sorter (BD Biosciences, Franklin Lakes, NJ, USA) and Sony LE-SH800 cell sorter (Sony Corporation, Tokyo, Japan). Human samples were processed using BD FACSAriaII cell sorter (BD Biosciences, Franklin Lakes, NJ, USA) and Sony LE-SH800 cell sorter (Sony Corporation, Tokyo, Japan). The projected cell surface area was measured using forward scatter area (FSC-A) signal. At least 10,000 cells were collected. The data were analyzed using FCS Express 6 (De Novo Software, Pasadena, CA, USA).

### 2.4. Membrane Potential Estimation

RBCs were washed three times with Ringer’s solution (154 mM NaCl, 2 mM KCl (Sigma-Aldrich, St. Louis, MO, USA)) prepared in-house. Packed RBCs were injected into Ringer’s solution supplemented with 20 µM protonophore CCCP [12] (Sigma-Aldrich, St. Louis, MO, USA) reaching 3.3% hematocrit and maintained at 37 °C. Three minutes after RBC addition, 100 µM of NS309 was added (a compound which shifts Ca^2+^ sensitivity of Gárdos of one order of magnitude [13] (Alomone labs, Jerusalem, Israel) followed by RBC lysis 8 min after RBC addition using 3M NaCl 1% Triton solution (Sigma-Aldrich, St. Louis, MO, USA). Extracellular pH (pH_out_) was measured every second by a pH electrode. Intracellular pH (pH_in_) remained constant throughout the experiment because of the high RBC buffering capacity. The transmembrane potential (V_m_) was calculated using the Nernst equation adjusted for RBCs, where V_m_ = 61.51 mV × (pH_in_ − pH_out_). Final lysis gives absolute values of membrane potential, since at that point, pH_in_ = pH_out_.

### 2.5. RBC Volume Assays

RBCs were washed three times with PBS (Thermo Fisher Scientific, Waltham, MA, USA), and 0.05% RBC suspension was prepared in PBS with a final concentration of 0.2% BSA and 1 mM CaCl_2_. RBC volume was measured using CASY system (Roche Innovatis, Bielefeld, Germany) before and 2.5, 5, 7.5, and 10 min after 100 μM NS309 addition.

### 2.6. Automated Patch-Clamp

Automated patch-clamp on the SyncroPatch 384PE (Nanion Technologies, Munich, Germany) was used to measure Gárdos activity, with a modified protocol previously described [14]. Briefly, after three washes with PBS and upon cell catch, external recording solution was added to the wells followed by 10 µM NS3623 (a powerful chloride channel inhibitor [15]), 10 µM NS309 (Alomone labs, Jerusalem, Israel), 5 µM TRAM-34 (a specific Gárdos channel inhibitor, Tocris, Bristol, UK), and 30 µM GdCl_3_ (a generic non selective cation channel inhibitor, Sigma-Aldrich, St. Louis, MO, USA). Internal solution (in mM): 10 KCl, 110 KF, 10 NaCl, 10 EGTA, and 10 HEPES/KOH (pH 7.2). External solution (in mM): 140 NaCl, 4 KCl, 2 CaCl_2_, 1 MgCl_2_, 5 D-Glucose monohydrate, and 10 HEPES/NaOH (pH 7.4) (Nanion Technologies, Munich, Germany). All compounds (NS3623, NS309, TRAM-34, and GdCl_3_) were diluted in an external solution. Currents were measured at room temperature, applying −100 to +80 mV ramp voltage protocol for 300 ms every 10 s at a holding potential of −30 mV. The cell response was measured in pA at +80 mV. Each cell underwent strict quality control (QC) filters, and only NS309 and TRAM-34 responder cells were used in the analysis (Appendix A).

### 2.7. Imaging Flow Cytometry

Mice bone marrow, spleen, and peripheral blood samples were washed three times with PBS (Thermo Fisher Scientific, Waltham, MA, USA). For total mitochondria detection, the samples were stained with 500 nM MitoFluor Red 589 dye (M-22424) (Molecular Probes, Eugene, OR, USA) in 120 µL PBS+0.5% BSA for 30 min at 37 °C with subsequent two washes with PBS+0.5% BSA. Bone marrow and spleen cells were stained with CD71-FITC conjugated antibody (BD Biosciences, Franklin Lakes, NJ, USA) at 1/50 dilution and Ter119-PE conjugated antibody (Biolegend, San Diego, CA, USA) at 1/25 dilution in 120 µL of PBS+0.5% BSA for 30 min at RT. After two washes with PBS+0.5% BSA bone marrow, spleen and peripheral blood samples were stained for nuclei using Hoechst 33342 dye (Life Technologies, Carlsbad, CA, USA) for 10 min at RT at 40 µg/mL in 70 µL PBS+0.5% BSA. Samples were acquired using ImageStream^®X^ Mk II Imaging Flow Cytometer (Merck Group, Darmstadt, Germany). At least 300,000 Ter119^+^ cells from the bone marrow and spleen and 100,000 cells from peripheral blood were collected. The data were analyzed using IDEAS 6.2 software (Merck Group, Darmstadt, Germany) with at least 100 single cell images in each subpopulation. The Ter119, CD71, Hoechst and MitoFluor signals were analyzed using mean fluorescence intensity, if not stated otherwise.

For nascent pyrenocyte and nascent reticulocyte analysis, we developed a gating strategy and masks based on Hoechst stain and brightfield (BF) images. First, elongated Ter119^+^/Hoechst^+^ cells were selected using the BF shape ratio range 0.45–0.70, and delta centroid Hoechst and BF range 0–4. From the elongated cell population, we selected enucleating cells using a cell area mask, where the nascent pyrenocyte area mask was defined as the area covered by Hoechst positive staining at a range of 25–50 µm^2^ and the nascent reticulocyte area mask was defined as a BF area without the Hoechst area at a range of 20–30 µm^2^.

To analyze the cell size area during in vivo erythropoiesis, we developed a gating strategy by selecting the cell-facing camera. We used BF object circularity using a tight cell size mask with a cut-off of 12.8. The cell area was measured in µm^2^. To compare the area change dynamics during in vivo erythroid differentiation, we subdivided bone marrow and spleen Ter119^+^ cells into the Hoechst^+^/CD71^+^, Hoechst^−^/CD71^+^, and Hoechst^-^/CD71^-^ subpopulations. Only Ter119^+^/Hoechst^-^ cells were analyzed from peripheral blood samples.

### 2.8. Sample Preparation for Tandem Mass Spectrometry (MS/MS) Analysis

Fresh peripheral blood samples from aspirin-treated or non-treated JAK2^V617F^ PV (*n* = 4) and ET (*n* = 4) patients, and healthy donors (*n* = 4) were used. The erythroid cells were selected by cellulose separation following density gradient separation enriching the samples in reticulocytes as previously described [7]. At least 1 × 10^6^ reticulocytes and mature RBCs were sorted using BD FACSAriaII cell sorter (BD Biosciences, Franklin Lakes, NJ, USA) and Sony LE-SH800 cell sorter (Sony Corporation, Tokyo, Japan). Samples were washed with PBS and immediately stored at −80 °C. Stored RBCs were then lysed for 5 min at 95 °C in 200 mM Triethylammonium bicarbonate (TEAB), pH 8.5, 2% sodium dodecyl sulfate. Protein concentrations were determined using a bicinchoninic acid assay (BCA kit, Pierce, Waltham, MA, USA). Proteins were reduced and alkylated in 10 mM TCEP ((tris(2-carboxyethyl)phosphine) and chloroacetamide. Thirty micrograms of proteins was digested overnight with trypsin using the suspension trapping (S-Trap) method (ProtiFi). Eluted peptides were then fractionated using strong cation exchange (SCX) StageTips in five fractions as previously described [16] and vacuum-dried while centrifuged in a Speed Vac (Eppendorf, Hamburg, Germany).

### 2.9. Nanoscale Liquid Chromatography Coupled to Tandem Mass Spectrometry Analysis

Mass spectrometry (MS) analyses were performed on a Dionex U3000 RSLC nano-LC system coupled to an Orbitrap Fusion mass spectrometer (Thermo Fisher Scientific, Waltham, MA, USA). Peptides from each SCX fraction were solubilized in 0.1% trifluoracetic acid containing 10% acetonitrile and separated on a C18 column (2 mm particle size, 75 mm inner diameter, 25 cm length; Thermo Fisher Scientific, Waltham, MA, USA) with a 3 h gradient starting from 99% solvent A (0.1% formic acid) and ending with 55% solvent B (80% acetonitrile, 0.085% formic acid). The mass spectrometer operated in a data-dependent manner with full MS scans acquired with the Orbitrap, followed by higher-energy collisional dissociation fragmentations of the most abundant ions with detection in the ion trap for 3 s.

### 2.10. Analyses of MS/MS Data

The MS data were analyzed using MaxQuant software version 1.6.6.0 (Max Planck Institute of Biochemistry, Martinsried, Germany) [17] using the Uniprot-Swissprot reviewed database (Uniprot-Swissprot, release 2019-10, European Bioinformatics Institute, Hinxton, UK; Swiss Institute of Bioinformatics, Lausanne, Switzerland; Protein Information Resource, Washington, DC, USA) and a protein false discovery rate of 0.01. Carbamidomethylation of cysteines was set as a constant modification and acetylation of protein N termini and oxidation of methionines as variable modifications. Labelfree protein quantification (LFQ) was performed using both unique and razor peptides with at least two ratio counts. Data from all experiments were analyzed simultaneously using the match between runs (MBR) option. Functional analyses of data from MaxQuant were conducted with Perseus software [18].

### 2.11. SDS-PAGE Immunoblots

BaF3 and HEL cell samples were lysed for 45 min at 4 °C in lysis buffer containing: 20 mM Tris, 150 mM NaCl, 5 mM EDTA, 0.002% NaN_3_, 1% Triton X-100, 0.2% BSA, phosphatase (Sigma-Aldrich, St. Louis, MO, USA), and protease inhibitor cocktails (Roche Diagnostics, Basel, Switzerland), and 50 µg of protein lysates containing Laemmli 1X and 5% β-mercaptoethanol (Sigma-Aldrich, St. Louis, MO, USA) were size-separated in 8% SDS-PAGE and transferred on nitrocellulose membrane (Whatman-Protran, Merck Group, Darmstadt,, Germany). Gárdos protein levels were detected by anti-K_Ca_3.1 antibody (Alomone labs, Jerusalem, Israel) and anti-mouse-HRP conjugated antibody (Abliance, Compiegne, France). Anti-β-actin-HRP conjugated antibody (Cell Signaling, Danvers, MA,, USA) served as a loading control. Signal was revealed using ECL substrate with ChemiDoc MP system and quantified using Image lab 6.0 (Biorad, Hercules, CA, USA).

### 2.12. Statistical Analysis

Mann–Whitney test or Wilcoxon test was used for statistical analysis using Prism 7 (GraphPad, San Diego, CA, USA), where * *p* < 0.05; ** *p* < 0.01; *** *p* < 0.001; **** *p* < 0.0001; ns—not significant. Data are presented as the mean value with standard deviation (SD), if not stated otherwise.

## 3. Results

### 3.1. PV RBCs Show Elevated Free Intracellular Ca^2+^ Levels and Increased Gárdos Channel Activity

We measured the free intracellular Ca^2+^ levels in RBCs from nine PV patients and nine healthy donors (CT) by flow cytometry using the Ca^2+^ tracer Fluo4-AM. In the absence of added extracellular CaCl_2_, Fluo4-positive cell count and mean fluorescence intensity (MFI) were higher in PV than in CT RBCs (Figure 1A,B).

To assess the altered intracellular calcium concentration and, thus, the calcium homeostasis, we used a functional assay based on the Ca^2+^ sensitivity of the Gárdos channel. Using NS309, a compound able to shift calcium sensitivity of KCNN4, relative Ca^2+^ intracellular concentration can be estimated by the extent of hyperpolarization once Gárdos channel is activated, since calcium acts directly on the open probability of the Gárdos channel. In this experiment, changes in the membrane potential were determined using the CCCP method. When packed RBCs were added (t = 30 s) to the Ringer’s solution supplemented with 20 µM protonophore CCCP, CT and PV RBCs displayed the expected resting membrane potential of −10 mV (Figure 2A). The subsequent addition of 100 µM of NS309 (t = 210 s) activates the Gárdos channel according to the nominal intracellular Ca^2+^ concentration, leading to a hyperpolarization. The amplitude of membrane hyperpolarization is more pronounced in PV cells compared with CT cells (Figure 2A). Such an observation suggests that the intracellular concentration of Ca^2+^ is increased in a fraction of PV RBCs. Activation of the Gárdos channel results in dehydration and shrinkage because of H_2_O loss following K^+^ and Cl^-^ efflux. We measured the changes in the RBC volume in the presence of 100 µM NS309 and 1 mM CaCl_2_. As expected, RBCs showed continuously decreased cell volume upon the addition of NS309 (Figure 2B). PV RBCs showed faster dynamics of cell volume loss compared with CT RBCs, with a significant greater decrease at 7.5 min (Figure 2B), supporting increased Gárdos activity in PV RBCs.

To confirm the affected calcium homeostasis in PV RBCs and to further explore the role of JAK2^V617F^, we measured the Gárdos activity at a single cell level using patch-clamp in RBCs and cell lines expressing JAK2^V617F^ or JAK2^WT^. We developed a patch-clamp protocol using SyncroPatch (Appendix A), a patch-clamp device used to study channel activities in RBCs in an automated high-throughput manner [14]. Gárdos activity was measured at baseline and in the presence of a Gárdos activator, NS309, and an inhibitor, TRAM-34, for PV and CT RBCs (Figure 2C left panel). PV RBCs (*n* = 48) showed significantly higher NS309-induced current than CT RBCs (*n* = 17) (Figure 2C middle panel). The TRAM-34-induced current change was also greater in PV RBCs compared with CT RBCs (Figure 2C right panel). These single-cell patch-clamp results further confirmed the increased Gárdos activity in PV RBCs in the absence of a different expression level of Gárdos between PV and CT RBCs, as determined by flow cytometry (Appendix A). To determine if the increased Gárdos channel activity is driven by JAK2^V617F^, we performed patch-clamp analyses using two murine pro-B BaF3 cell lines carrying recombinant human JAK2^WT^ or JAK2^V617F^ as well as the human erythroleukemia cell line HEL expressing endogenous JAK2^V617F^ in the presence or absence of ruxolitinib, a JAK2 inhibitor. Gárdos channel activity in the presence of NS309 was significantly increased in BaF3 JAK2^V617F^ cells (*n* = 48) compared with in BaF3 JAK2^WT^ (*n* = 20) (Figure 2D left and middle panels). Inhibition of the Gárdos channel by TRAM-34 resulted in increased current change in BaF3 JAK2^V617F^ cells compared with BaF3 JAK2^WT^ cells (Figure 2D right panel). Similarly, HEL cells without ruxolitinib treatment (*n* = 61) showed higher Gárdos channel activity upon NS309 addition compared with HEL cells treated with ruxolitinib (*n* = 63) (Figure 2E left and middle panels). Inhibition of the Gárdos channel resulted in a higher current change in HEL cells compared with those incubated with ruxolitinib (Figure 2E right panel). These patch-clamp results indicated an increased Gárdos channel activity in a JAK2^V617F^ context in the absence of the differential expression of Gárdos at the protein level among the cell lines and conditions (Supplemental Appendix A), suggesting a relationship between JAK2^V617F^ and increased intracellular Ca^2+^ levels.

### 3.2. JAK2^V617F^ Is Associated with Decreased Erythroid Cell Size during Mouse In Vivo Erythropoiesis

PV RBCs are known to be smaller than normal RBCs, and it has been suggested that insufficient iron levels are the cause of this microcytosis [19]. Our calcium data suggest that this microcytosis can also be the consequence of mild cell dehydration in a JAK2^V617F^ context. To address this hypothesis, we measured the size of peripheral blood mature RBCs and reticulocytes using flow cytometry forward scattered area (FCS-A) in a JAK2^WT^ and JAK2^V617F^ knock-in mouse model in which lower mean corpuscular volume (MCV) has been reported for JAK2^V617F^ than for JAK2^WT^ RBCs [8]. As expected, JAK2^V617F^ mice had smaller mature RBCs and reticulocytes compared with JAK2^WT^ mice (Figure 3A,B). JAK2^V617F^ mature RBCs and reticulocytes had mean area ratios of 0.78 (SD = 0.16) and 0.81 (SD = 0.10), respectively, when normalized to their JAK2^WT^ counterpart. We further confirmed these results by performing cell size measurements using imaging flow cytometry. We developed a mask solely selecting cells facing the camera and those with brightfield object circularity higher than 12.8 (Figure 3C). In accordance with the flow cytometry data, the JAK2^V617F^ mature RBCs showed a lower projected surface area compared with JAK2^WT^ RBCs (Figure 3D). To address the potential role of JAK2^V617F^ in this reduced cell size, we analyzed erythroid cells during in vivo erythroid differentiation. Cells were collected from mouse bone marrow and spleen and stained for Ter119 (erythroid cells) and with Hoechst (DNA dye), MitoFluor (mitochondria dye), and an anti-CD71 (transferrin receptor) antibody to distinguish between nucleated erythroblasts (Hoechst^+^/CD71^+^/MitoFluor^+^), early reticulocytes (Hoechst^-^/CD71^+^/MitoFluor^+^), and late reticulocytes (Hoechst^-^/CD71^-^/MitoFluor^+^). As expected, there was a loss in the cell size, as determined by the projected surface area, when progressing from the nucleated erythroblast to the late reticulocyte stage, in both bone marrow (Figure 3E) and spleen (Figure 3F). In bone marrow, erythroblasts displayed no difference in the cell size between JAK2^WT^ and JAK2^V617F^ mice, but JAK2^V617F^ early reticulocytes were smaller than those from JAK2^WT^ mice (Figure 3E), suggesting potential alterations during the enucleation step. Nevertheless, the JAK2^WT^ and JAK2^V617F^ late reticulocytes showed similar cell surface area (Figure 3E), indicating normalization of the erythroid cell size in JAK2^V617F^ mice during terminal maturation. In the spleen, there was a difference in cell size at all three stages, with JAK2^V617F^ cells showing smaller surface area than JAK2^WT^ cells (Figure 3F), suggesting alterations during terminal stress erythropoiesis in a JAK2^V617F^ context. Furthermore, late reticulocytes showed smaller surface area (µm^2^) in the spleen than in the bone marrow (50.4 ± 1.0 and 54.0 ± 2.5, respectively; *p* = 0.0152), suggesting that the majority of circulating mature RBCs in JAK2^V617F^ mice are generated by extramedullary stress erythropoiesis.

### 3.3. JAK2^V617F^ Is Associated with Higher Organelle Remnants in Circulating Mouse Reticulocytes

To gain insight into the role of JAK2^V617F^ during terminal erythroid differentiation, we investigated organelle sorting during the enucleation step. We determined the ribosomal content in circulating reticulocytes by staining the ribosomal RNA with thiazole orange (TO) (Figure 4A) and by measuring the mean fluorescence intensity of TO-positive JAK2^WT^ and JAK2^V617F^ reticulocytes by flow cytometry (Figure 4B). We determined the RNA levels in reticulocytes from seven mice in each group and showed higher percentages of TO^+^ circulating erythrocytes in JAK2^V617F^ than in JAK2^WT^ mice (Figure 4C). Furthermore, we found a 1.29-fold increase in TO MFI in JAK2^V617^ mice compared with JAK2^WT^ (Figure 4D), indicating more ribosomes in JAK2^V617^ reticulocytes. To check if this was restricted to ribosomes, we stained cells for the presence of mitochondria using MitoFluor and analyzed them using imaging flow cytometry. JAK2^WT^ and JAK2^V617F^ mouse peripheral blood was stained with MitoFluor and Hoechst to allow for the identification and the exclusion of nucleated cells (Figure 4E). JAK2^V617^ mice showed higher percentages of Hoechst^-^MitoFluor^+^ circulating reticulocytes (Figure 4F) and higher amounts of mitochondria within these cells as estimated by the mean fluorescence intensity of Hoechst^-^MitoFluor^+^ cells (Figure 4G).

### 3.4. Altered Organelle Sorting during Enucleation of JAK2^V617F^ Mouse Erythroblasts

The increased amounts of ribosomes and mitochondria in circulating JAK2^V617F^ reticulocytes suggests potential alteration during the erythroblast enucleation step (Figure 5A upper panel). To explore this hypothesis, we stained differentiating erythroid cells from bone marrow and spleen of JAK2^WT^ and JAK2^V617F^ mice. We divided Ter119^+^ (erythroid) cells into five sub-populations according to the cell maturity stage based on the presence or absence of nucleus and the expression levels of CD71 set as high, medium, low, and negative and using MitoFluor dye as an organelle marker (Figure 5A bottom panel). As expected, bone marrow nucleated cells had higher mitochondria levels compared with enucleated cells (Figure 5B upper panel). There were similar mitochondria levels in nucleated erythroid bone marrow cells between JAK2^WT^ and JAK2^V617F^ mice (Figure 5B upper panel), indicating that JAK2^V617F^ does not drive higher amounts of mitochondria in these cells. However, JAK2^V617F^ early reticulocytes (Hoechst^-^/CD71^high^ cells) showed higher mitochondria levels than JAK2^WT^ cells (Figure 5B upper panel), suggesting altered organelle sorting at this early stage. Nevertheless, as the reticulocytes progressed to more mature stages (CD71^med, low and neg^ cells), the difference in mitochondria levels between JAK2^WT^ and JAK2^V617F^ mice was lost (Figure 5B upper panel). Similar to the bone marrow erythropoiesis, spleen nucleated erythroid cells from JAK2^WT^ and JAK2^V617F^ mice displayed similar mitochondria levels (Figure 5B bottom panel), with the expected decrease in mitochondria levels after enucleation (Figure 5B bottom panel). Spleen early reticulocytes (Hoechst^-^/CD71^high^ cells) also showed elevated mitochondria levels compared with JAK2^WT^ cells (Figure 5B bottom panel). Opposite to the bone marrow, during stress erythropoiesis in the spleen, the increased mitochondria levels were also present at later stages in JAK2^V617F^ CD71^med and low^ reticulocytes compared with JAK2^WT^ cells (Figure 5B bottom panel). Although not statistically significant, JAK2^V617F^ late reticulocytes and mature RBCs (CD71^-^ cells) showed a trend with increased mitochondria levels compared with JAK2^WT^ cells (Figure 5B bottom panel, *p* = 0.0931). Interestingly, enucleated JAK2^WT^ cells displayed similar mitochondria levels throughout reticulocyte maturation, while JAK2^V617F^ cells displayed continuous decrease in these levels during bone marrow and spleen erythropoiesis (Figure 5B). Due to the very low mitochondria levels in reticulocytes, the observed MitoFluor signal in enucleated JAK2^WT^ cells was considered as non-specific background.

In order to finely analyze organelle sorting during the enucleation step, we developed masks and gating strategies using imaging flow cytometry to assess MitoFluor intensity distribution in mouse nascent pyrenocytes and nascent reticulocytes before the completion of enucleation. First, we selected elongated Ter119^+^/Hoechst^+^ (nucleated erythroid) cells using the brightfield (BF) shape ratio and Hoechst and BF delta centroid features (Figure 6A left panel). We then selected enucleating cells from the elongated cell population based on the projected area size of nascent pyrenocytes (Hoechst^+^ area) and nascent reticulocytes (Hoechst^-^ area) (Figure 6A middle panel). Figure 6A right panel shows a representative example of a selected cell from the enucleating cell population. We measured MitoFluor intensity in the areas defined by the previously used masks as nascent reticulocytes and nascent pyrenocytes (Figure 6B). As expected, nascent reticulocytes showed lower MitoFluor intensity compared with nascent pyrenocytes and with the whole enucleating erythroblast (Figure 6C). JAK2^V617F^ nascent pyrenocytes showed lower MitoFluor intensity compared with JAK2^WT^ in both bone marrow and spleen (Figure 6D). At the contrary, JAK2^V617F^ nascent reticulocytes showed higher MitoFluor intensity compared with JAK2^WT^ in both tissues (Figure 6D), supporting the data of Figure 5B showing higher MitoFluor intensity in bone marrow and spleen reticulocytes and indicating altered organelle sorting in a JAK2^V617F^ context.

### 3.5. Higher Ribosomal Content and Smaller Red Cell Size in JAK2^V617F^ PV and ET Patients

To extend these data to human cells, we analyzed circulating reticulocytes and mature RBCs from patients with PV and ET carrying the JAK2^V617F^ mutation and healthy individuals (CT). Cells were analyzed for organelle remnants as well as for their size. Due to the very low percentages of circulating reticulocytes, we applied an enrichment protocol based on cellulose separation of blood samples followed by reticulocyte enrichment using a cell density gradient, as previously described [7] (Figure 7A). Reticulocytes from PV patients showed a 1.32-fold increase in ribosomal RNA levels compared with CT (Figure 7B,C). Similar to the mouse results, human PV RBCs and reticulocytes were smaller compared with CT RBCs (Figure 7D,E). When normalized to the area of CT cells (*n* = 9), PV mature RBCs and reticulocytes (*n* = 4) showed mean area ratios of 0.85 (SD = 0.12) and 0.86 (SD = 0.11), respectively (Figure 7F,G). Interestingly, cells from ET JAK2^V617F^ patients (*n* = 10) also had significantly lower cell areas than CT (*n* = 9), reaching 0.92 (SD = 0.05) in mature RBCs (Figure 7F) and 0.95 (SD = 0.04) in reticulocytes (Figure 7G), with no significant difference with PV cells (*n* = 4). Finally, we measured the projected RBC surface in PV and ET patients using imaging flow cytometry. Cells facing the camera were selected based on circularity (cutoff 12.1) and max thickness features (cutoff 8.1). JAK2^V617F^ PV (*n* = 7) and ET RBCs (*n* = 5) were significantly smaller than CT RBCs (*n* = 15) (Figure 7H).

### 3.6. Modified Whole Cell Proteome in Circulating Reticulocytes and Mature RBCs from PV and ET Patients with JAK2^V617F^

To validate the role of JAK2^V617F^ in protein and organelle missorting and reticulocyte maturation, we used a whole cell proteomics approach. We quantified cell proteome by performing nanoscale liquid chromatography coupled to tandem mass spectrometry (MS/MS) analysis of sorted mature red blood cells and reticulocytes from four PV, four ET patients with JAK2^V617F^ mutation, and four healthy individuals (CT); 1265 proteins were selected after filtering out reverse proteins, proteins that were only identified by site, contaminant proteins, and proteins with less than three valid values in at least one group (Appendix A). Each comparative group identified a number of proteins with altered levels (range 21–82) using a cutoff fold ratio of >1.2 (Table 1 and Appendix A) and a range of on/off proteins between groups (1–215) (Table 1, Appendix A). We found 11 more abundant proteins and 10 less abundant proteins in mature RBCs from JAK2^V617F^ PV patients compared with healthy individuals (Table 1). We defined protein cellular localization using UniProt annotation and GO–cellular component data. As expected, these proteins were mostly residing in cytoplasm or plasma membrane (Table 2). Interestingly, PV RBCs showed an eight-fold increase in MYH10 translated protein—a Ca^2+^ regulated non-muscle myosin IIB (NMIIB). The increase in 14-3-3 protein beta/alpha in PV RBCs compared with healthy individuals is consistent with our previous proteomics data from RBC membranes [5] and further validates our results for this whole cell proteomic study (Table 2). Additionally, we identified 39 more abundant proteins and 11 less abundant proteins in reticulocytes from JAK2^V617F^ PV patients compared with healthy individuals (Table 1). Reticulocytes from JAK2^V617F^ PV patients showed increased levels of multiple mitochondria, ribosomal, and ER residing proteins, such as voltage-dependent anion-selective channel protein 3 (Vdac3), 40S/60S ribosomal proteins, and protein disulfide-isomerase (Pdi), respectively, which is consistent with our imaging and flow cytometry data. Furthermore, reticulocytes from JAK2^V617F^ PV patients showed modified levels of multiple proteins residing in the cytoplasm, plasma membrane, nucleus, Golgi apparatus, endosomes, and cytoskeleton compared with reticulocytes from healthy individuals, suggesting alterations during erythroid differentiation and reticulocyte maturation in a JAK2^V617F^ context (Table 3 and Appendix A).

## 4. Discussion

Our study shows altered organelle sorting during the enucleation step of erythroid precursors harboring the JAK2^V617F^ mutation, resulting in higher amounts of organelles in nascent reticulocytes and subsequent functional alterations such as disturbed calcium homeostasis in circulating RBCs of PV patients.

JAK2 and Ca^2+^-dependent signaling regulates numerous pathways during enucleation and reticulocyte maturation. The organelle missorting during the enucleation step of JAK2^V617F^ erythroid precursors suggests that a continuous non-controlled signaling of the EpoR/JAK2 pathway drives abnormalities beyond the well-described proliferation triggered by the V617F mutation. Several genes involved in the enucleation process are regulated in a JAK2-dependent manner. One of these genes is c-MYC, in which the expression is rapidly induced upon Epo stimulation of committed erythroid progenitor cells [20] and reduced during the final stages of erythroid maturation [21]. This downregulation is essential for terminal erythroid maturation, as the continuous expression of c-MYC at physiological rates blocks nuclear condensation and enucleation [22]. Activated JAK2 increases c-MYC gene transcription and interferers with c-Myc protein degradation, leading to higher c-MYC cellular levels [23], which would negatively impact the enucleation process of JAK2^V617F^ erythroid precursors. Another candidate protein that may interfere with the enucleation process is Trim 58, which regulates nuclear polarization through dynein degradation in mouse and human erythroid precursors [24,25]. Trim 58 levels are increased in PV JAK2^V617F^ erythrocytes compared with those from healthy individuals [26], suggesting that JAK2^V617F^ may alter microtubule-mediated nuclear polarization.

Formation of contractile actin ring between nascent pyrenocyte and reticulocyte by Ras-activated mDia2 polymerization of actin filaments is essential for enucleation [27,28]. Even though mDia2 is not a target of JAK2, it is worthy to note that the greatest changes in PV JAK2^V617F^ erythrocyte and granulocyte protein expression belong to members of the RAS superfamily of GTPase signaling pathways [26,29]. Similar to mDia2, the Ca^2+^ regulated non-muscle myosin IIB (NMIIB) interacts with actin, and its inhibition blocks enucleation in mouse and human erythroblasts [30,31]. MYH10 translated protein NMIIB is highly increased in PV RBCs, making it an attractive target for further investigation in organelle missorting events in PV. It is interesting to note that the switch between stress and steady erythropoiesis is regulated by the transport of the glucocorticoid receptor (GR) from the nucleus to the cytoplasm via Calr [32,33] and that Ca^2+^ regulates Calr conformation for this process, suggesting a role for calcium in stress erythropoiesis in PV [34].

A recent study showed that mDia2-deficient mice have a blockade of enucleation and a failure of organelle (mitochondria) clearance in reticulocytes [35]. PV patients show elevated levels of transcription factor nuclear factor-erythroid 2 (NF-E2) [36]. Autophagy genes NIX and ULK1 are direct targets of NF-E2, and mice overexpressing NF-E2 impairs mitochondria autophagy in reticulocytes [37]. Furthermore, Ca^2+^ regulated non-muscle myosin IIA (NMIIA) regulates vesicle transport during reticulocyte maturation [38]. Increased levels of the MYH9 gene translated protein NMIIA in reticulocytes from PV patients makes NMIIA as a good target for vesicle transport studies in a JAK2^V617F^ context. Autophagy is affected in multiple diseases during reticulocyte maturation, including β-thalassemia [39,40,41] and sickle cell disease (SCD) [42,43]. It has been demonstrated that autophagy modulators are beneficial in both SCD and β-thalassemia patients [44,45]. The impact of altered autophagy in JAK2^V617F^ context remains to be investigated.

The JAK2^V617F^-triggered raise in free intracellular Ca^2+^ levels could be a result of deregulation of Ca^2+^ channels. There are currently five known Ca^2+^ and non-selective cation channels on human RBCs: Ca_V_2.1 [46], transient receptor potential channel C6 (TRPC6) [47], Piezo1 [48], N-methyl-D-aspartate (NMDA) receptor [49], and Transient Receptor Potential Vanilloid Type 2 (TRPV2) [50]. Our proteomics study showed no difference in Piezo1 expression levels between PV and CT RBCs [5]. The absence of detection of the four other transporters supports their expression in low copy numbers [46,47,48]. The only classified Ca^2+^ exporter on the human RBC membrane is the ubiquitously expressed plasma membrane calcium ATPase (PMCA) [51,52]. ATPase plasma membrane Ca^2+^ transporting 4 (PMCA4B) levels are reduced by 50% in PV RBC membranes compared with healthy individuals [5], suggesting slower Ca^2+^ export. Interestingly, PMCA4B haplotypes in healthy individuals have been linked to reduced PMCA levels, resulting in reduced mean corpuscular hemoglobin concentrations (MCHC) [53]. Microcytosis in PV JAK2^V617F^ patients is associated with insufficient iron stores to compensate the increase in hematocrit [19]. However, not all of the PV JAK2^V617F^ patients exhibit increased hematocrit and increased hemoglobin values, and both of these parameters of ET JAK2^V617F^ patients are in the levels of healthy individuals [54]. JAK2^V617F^-affected Ca^2+^ homeostasis leads to increased Gárdos (K_Ca_3.1) activity and subsequent cell dehydration, making K_Ca_3.1 a potential target for further microcytosis investigations.

Thrombotic events are the main cause of mortality in PV patients. The presence of organelle remnants in circulating erythrocytes could affect the properties of the cells and potentially contribute to thrombotic event formation in PV. Furthermore, increased blood viscosity in PV patients has an effect on the rheological properties, which can be further modified by Ca^2+^ [55]. Increased Ca^2+^ levels are known to trigger intercellular adhesion of RBCs [56,57,58]. Moreover, Ca^2+^ facilitates endothelium–RBC adhesion [59,60,61,62]. Future studies in PV should investigate the role of Ca^2+^ in cell adhesion of more cell types carrying JAK2^V617F^, such as neutrophils, platelets, and endothelial cells.

## 5. Conclusions

Our study shows modified calcium homeostasis in PV RBCs and cells expressing JAK2^V617F^, with a functional impact on the activity of the Gárdos channel that could contribute to cellular dehydration. Moreover, we show that JAK2^V617F^ affects organelle sorting during enucleation and reticulocyte maturation. Our study opens new perspectives to exploring the relationship between JAK2^V617F^, calcium homeostasis, and cellular abnormalities in MPNs, including cellular interactions in the bloodstream in relation to thrombotic events.

## Figures and Tables

**Figure 1 cells-11-00049-f001:**
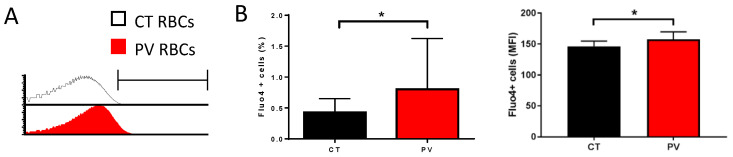
PV RBCs show elevated free intracellular Ca^2+^ levels. (**A**) Representative histogram overlays from Fluo4 stained CT (healthy individual) and PV RBCs in an isotonic solution without extracellular Ca^2+^. (**B**) Percentage of Fluo4-positive RBCs (**left** panel) and mean fluorescence intensity (MFI) of the positive population (**right** panel) without extracellular Ca^2+^. Mean with SD, *n* = 9, Wilcoxon test, * *p* < 0.05.

**Figure 2 cells-11-00049-f002:**
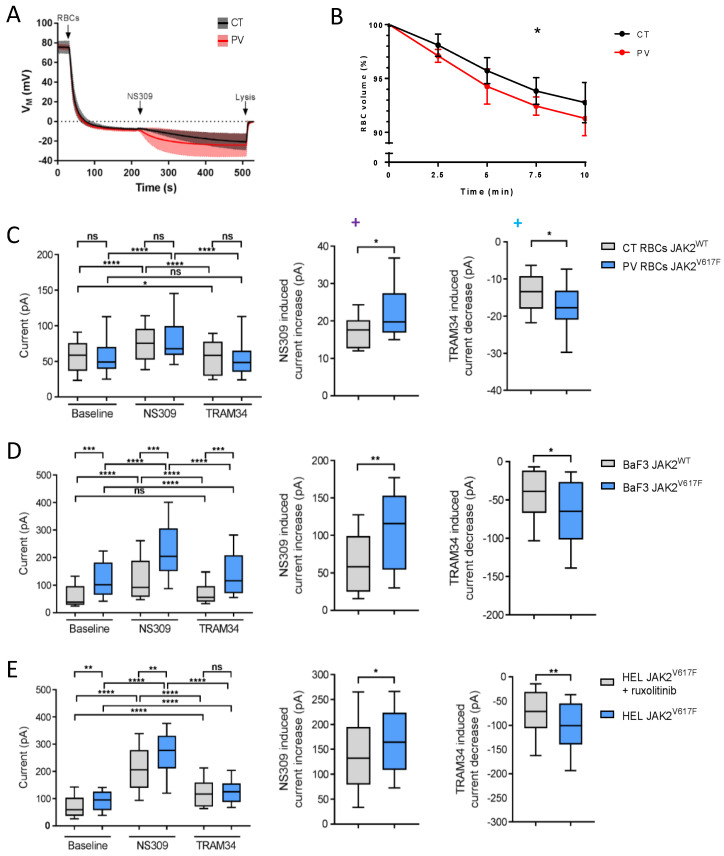
PV RBCs show increased Gárdos channel activity. (**A**) CCCP method analysis of PV and CT RBC membrane potential changes upon 100 µM NS309 addition. At the end of the experiment, cells were lysed with 3M NaCl 1% Triton X lysis solution to obtain the zero membrane potential (pHi = pHo) for absolute calculation of membrane potential. CT (mean—black line; SD—grey) and PV (mean—red line; SD—pink) RBCs. Data are displayed as mean with 95% confidence interval; CT (*n* = 6) and PV (*n* = 8). (**B**) Cell volume assay on Gárdos activity; 0.05% RBCs suspension was prepared in PBS with a final concertation of 0.2% BSA, 1 mM CaCl_2_, and 100 µM NS309. RBC size was measured using CASY before and 2.5, 5, 7.5, and 10 min after 100 μM NS309 addition. Mean with SD, *n* = 6, Mann–Whitney test. (**C**–**E**) Patch-clamp analysis. Upon cell catch external solution was added to the wells followed by 10 µM NS3623, 10 µM NS309, 5 µM TRAM-34, and 30 µM GdCl_3_. Currents were measured at room temperature applying −100 to +80 mV ramp voltage protocol for 300 ms, at a holding potential of −30 mV. The cell response was measured in pA at +80 mV. Statistical analysis of the currents at +80 mV in NS309 and TRAM-34 responding cells in (**C**) CT and PV RBCs (*n* = 17; *n* = 48, respectively), (**D**) BaF3 EpoR JAK2^WT^ (*n* = 20) and BaF3 EpoR JAK2^V617F^ (*n* = 48), and (**E**) HEL (*n* = 61) and HEL cells treated with 0.3 µM ruxolitinib for 24 h (*n* = 63). Cell was considered responsive if it displayed at least a 20% current change. The left panel represents the cell current upon the addition of NS3623 (baseline), NS309, and TRAM-34. The central panel displays NS309-induced current increase, while the right panels represent TRAM-34-induced current decrease. The data are presented as median and box plots (25–75%) with whiskers (10–90%). Mann–Whitney test or Wilcoxon test, * *p* < 0.05; ** *p* < 0.01; *** *p* < 0.001; **** *p* < 0.0001; ns—not significant.

**Figure 3 cells-11-00049-f003:**
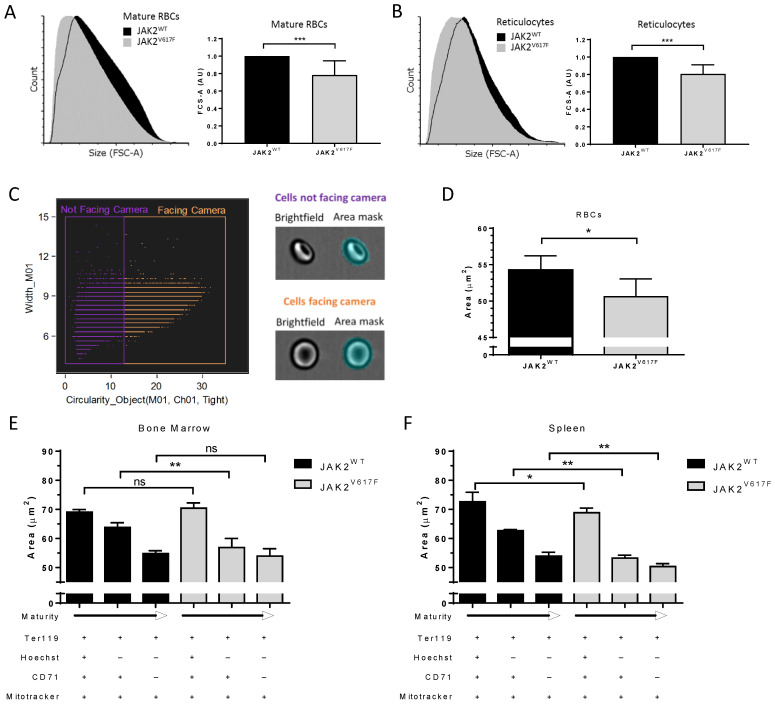
JAK2^V617F^ is associated with decreased erythroid cell size during mouse in vivo erythropoiesis. Representative forward scatter area (FCS-A) flow cytometry histogram of (**A**) (**left** panel) mature RBCs and (**B**) (**left** panel) reticulocytes from JAK2^WT^ and JAK2^V617F^ mice peripheral blood. Analysis of FCS-A (cell size) in (**A**) (**right** panel) mature RBCs and (**B**) (**right** panel) reticulocytes from JAK2^WT^ and JAK2^V617F^ mice peripheral blood. *n* = 7, mean with SD, Mann–Whitney test. (**C**) (**left** panel) Gating strategy and (**right** panel) representative images selecting the cells facing the camera (orange) out from the cells not facing the camera (purple) by imaging flow cytometry. M01–cell area mask (*y*-axis), cut-off of 12.8 on Ch01 (brightfield) object circularity using a tight cell size mask (*x*-axis). (**D**) Analysis of Hoechst ^neg^/CD71^neg^ (mature RBC) area of JAK2^WT^ and JAK2^V617F^ mice peripheral blood cells facing the camera. *n* = 6, mean with SD, Mann–Whitney test. Analysis of erythroid cell size area changes during terminal erythroid differentiation in vivo in mice Hoechst^pos^/CD71^pos^ and Hoechst^neg^/CD71^pos or neg^ populations of the cells facing the camera in (**E**) bone marrow and (**F**) spleen. *n* = 6, mean with SD, Mann–Whitney test, * *p* < 0.05; ** *p* < 0.01; *** *p* < 0.001; ns—not significant.

**Figure 4 cells-11-00049-f004:**
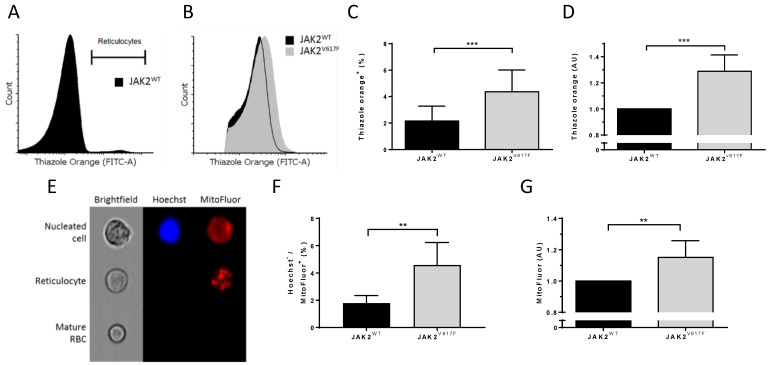
JAK2^V617F^ is associated with higher organelle remnants in circulating mouse reticulocytes. (**A**) Representative flow cytometry histogram gating strategy selecting the thiazole orange-positive (reticulocyte) population in JAK2^WT^ mouse peripheral blood. (**B**) Representative flow cytometry histograms of thiazole orange-positive populations between JAK2^WT^ and JAK2^V617F^ circulating reticulocytes. (**C**) Analysis of a thiazole orange-positive (reticulocyte) population in JAK2^WT^ and JAK2^V617F^ mouse peripheral blood. (**D**) Analysis of mean fluorescence intensity (MFI) of thiazole orange in circulating reticulocytes between JAK2^WT^ and JAK2^V617F^ mice. (**C**,**D**) *n* = 7, mean with SD, Mann–Whitney test. (**E**) Representative imaging flow cytometry example of a nucleated cell, reticulocyte, and a mature RBC from JAK2^WT^ mouse peripheral blood stained with Hoechst and MitoFluor dyes. (**F**) Percentage of Hoechst-negative and MitoFluor-positive population (reticulocytes) in JAK2^WT^ (*n* = 6) and JAK2^V617F^ mice (*n* = 5) peripheral blood. Mean with SD, Mann–Whitney test. (**G**) Analysis of MFI of MitoFluor in JAK2^WT^ (*n* = 6) and JAK2^V617F^ mice (*n* = 5) circulating reticulocytes. Mean with SD, Mann–Whitney test, ** *p* < 0.01; *** *p* < 0.001.

**Figure 5 cells-11-00049-f005:**
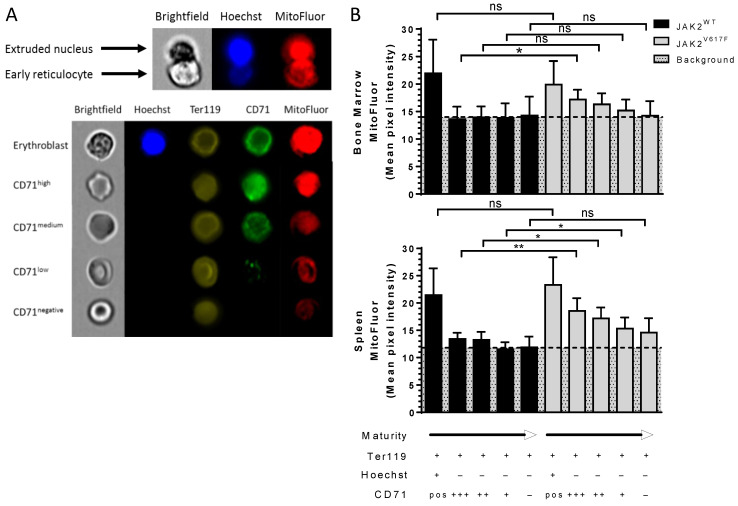
Altered organelle sorting during enucleation of JAK2^V617F^ mouse erythroblasts. Representative imaging flow cytometry images of (**A**) an enucleating orthoerythroblast and (**B**) an erythroblast and CD71^high, medium, low, negative^ expressing reticulocytes in mice stained with Hoechst, Ter119, CD71, and MitoFluor. Analysis of MitoFluor mean pixel intensity in erythroblasts and CD71^high, medium, low, negative^ expressing reticulocytes in JAK2^WT^ and JAK2^V617F^ mice; (**B**) (**top** panel) bone marrow and (**bottom** panel) spleen. *n* = 6, mean with SD, Mann–Whitney test, * *p* < 0.05; ** *p* < 0.01; ns—not significant.

**Figure 6 cells-11-00049-f006:**
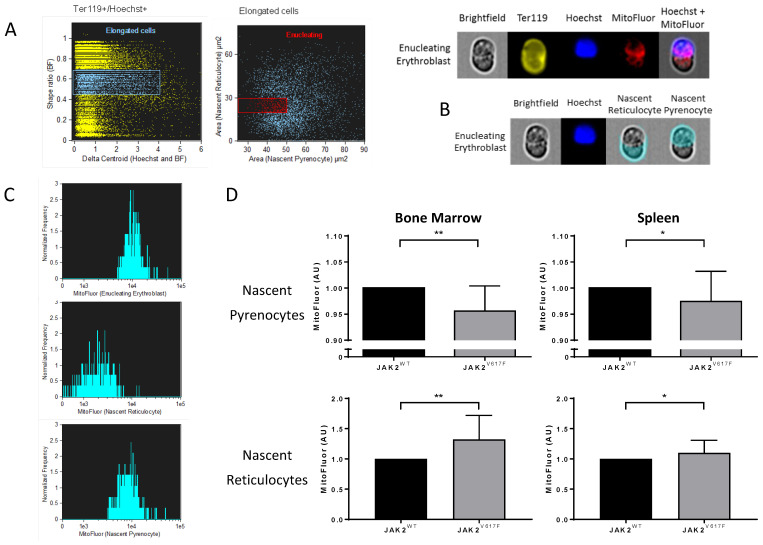
Altered organelle sorting during enucleation of JAK2^V617F^ mouse erythroblasts. (**A**) Gating strategy selecting mice enucleating erythroblasts by imaging flow cytometry. (**Left**) Selection of elongated Ter119^+^/Hoechst^+^ cells by a brightfield (BF) shape ratio range of 0.45–0.70, and a delta centroid Hoechst and BF range of 0–4. (**Right**) Enucleating cell selection from previously selected elongated cells by cell surface area. Nascent pyrenocyte area (*x*-axis) mask defined as the area covered by Hoechst positive staining, range 25–50 µm^2^. Nascent reticulocyte area (*y*-axis) mask defined as BF area without Hoechst area, range 20–30 µm^2^. (**Bottom** panel) Representative images of an enucleating erythroblast gated cell in BF, Ter119, Hoechst, MitoFluor, and Hoechst + MitoFluor images. (**B**) Representative area masks of an enucleating erythroblast developed for MitoFluor intensity measurements in nascent reticulocytes and nascent pyrenocytes. (**C**) Representative MitoFluor intensity histograms of enucleating erythroblasts (**top**), nascent reticulocytes (**centre**), and nascent pyrenocytes (**bottom**). (**D**) Analysis of MitoFluor intensity in JAK2^V617F^ cells normalized by the MitoFluor intensity of JAK2^WT^ cells. MitoFluor intensity in bone marrow nascent pyrenocytes (**top left**), spleen nascent pyrenocytes (**top right**), bone marrow nascent reticulocytes (**bottom left**), and spleen nascent reticulocytes (**bottom right**). *n* = 6, mean with SD, Mann–Whitney test, * *p* < 0.05; ** *p* < 0.01.

**Figure 7 cells-11-00049-f007:**
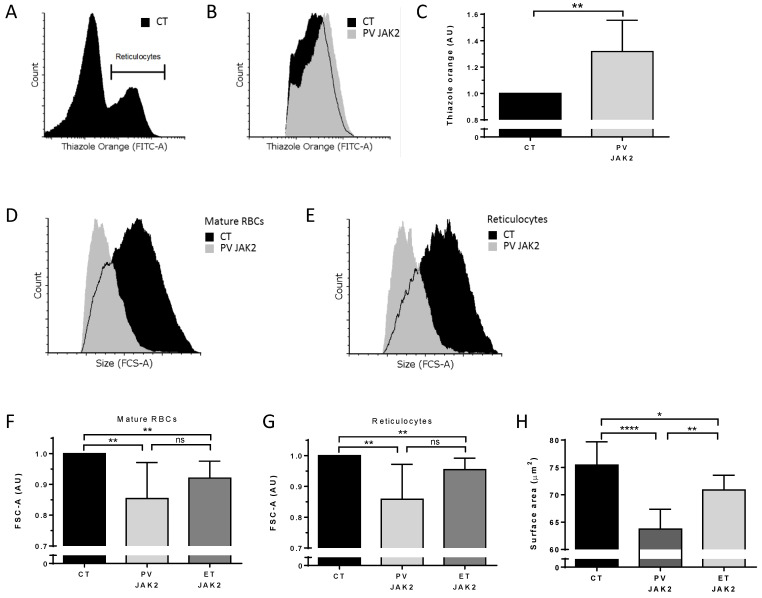
Higher ribosomal content and smaller red cell size in JAK2^V617F^ PV and ET patients. (**A**) Representative flow cytometry histogram gating strategy selecting a thiazole orange-positive (reticulocyte) population in a reticulocyte-enriched population from peripheral blood of a healthy individual (CT). (**B**) Representative flow cytometry histogram of thiazole orange-positive populations between CT and PV JAK2^V617F^ patient circulating reticulocytes. (**C**) Analysis of thiazole orange MFI in circulating reticulocytes from healthy individuals (*n* = 9) and non-treated PV JAK2^V617F^ (*n* = 4), ET JAK2^V617F^ (*n* = 10). Mean with SD, Mann–Whitney test. Representative forward scatter area (FCS-A) flow cytometry histogram of (**D**) mature RBCs and (**E**) reticulocytes from the peripheral blood of a healthy individual (CT) and a PV JAK2^V617F^ patient. Analysis of FCS-A (cell size) in (**F**) mature RBCs and (**G**) reticulocytes from peripheral blood of CT (N = 9) and non-treated PV JAK2^V617F^ (*n* = 4), ET JAK2^V617F^ (*n* = 10). Mean with SD, Mann–Whitney test. (**H**) Projected RBC surface in PV and ET patients using imaging flow cytometry. Cells facing the camera were selected based on circularity (cutoff 12.1) and max thickness features (cutoff 8.1). JAK2^V617F^ PV (*n* = 7), ET RBCs (*n* = 5), and CT RBCs (*n* = 15) mean with SD, Mann–Whitney test, * *p* < 0.05; ** *p* < 0.01; **** *p* < 0.0001; ns—not significant.

**Table 1 cells-11-00049-t001:** Number of quantified proteins that are significantly more or less abundant between groups, and number of on/off proteins between groups. Density gradient was used to enrich reticulocyte count from fresh human blood samples following mature red blood and reticulocyte sorting. Whole cell proteome was analyzed using nanoscale liquid chromatography coupled to tandem mass spectrometry (MS/MS). Significantly more or less abundant proteins between groups were analyzed using Student’s T-test *p* < 0.05 with protein level fold change cutoff >1.2; *n* = 4. On/off protein was defined as a protein detected in at least three samples out of four per group one, and no more than one sample out of four in group two. Polycythemia Vera (PV), essential thrombocythemia (ET), healthy individuals (CT), and V617F mutation in JAK2 (JAK2).

Cells	Comparison	More Abundant in X Group	More Abundant in Y Group	On Group X Off Group Y	On Group Y Off Group X
	X vs. Y				
Red Blood Cells	PV JAK2 vs. CT	11	10	14	23
	ET JAK2 vs. CT	12	18	11	24
	PV JAK2 vs. ET JAK2	13	12	14	5
Reticulocytes	PV JAK2 vs. CT	39	11	83	5
	ET JAK2 vs. CT	55	14	215	1
	PV JAK2 vs. ET JAK2	33	49	6	117

**Table 2 cells-11-00049-t002:** More and less abundant proteins in circulating mature red blood cells of PV JAK2^V617F^ patients compared with healthy individuals. Density gradient was used to enrich reticulocyte count from fresh human blood samples following mature red blood and reticulocyte sorting. Whole cell proteome was analyzed using nanoscale liquid chromatography coupled to tandem mass spectrometry (MS/MS). Significantly more or less abundant proteins between groups were analyzed using Student’s *t*-test *p* < 0.05 with protein level fold change cutoff >1.2; *n* = 4. Polycythemia Vera (PV), healthy individuals (CT), and V617F mutation in JAK2 (JAK2). Protein cellular localization was defined using UniProt annotation and GO–cellular component data. Subcellular locations: cytoplasm (C), plasma membrane (Pm), mitochondria (M), nucleus (N), cytoskeleton (S), endoplasmic reticulum (Er), and endosome (E).

Red Blood CellsPV JAK2 vs. CT	GeneName	AccessionNumber	*p*-Value	Fold Change	Subcellular Location
Protein	Cytoplasm	Plasma Membrane	Mitochondria	Nucleus	Cytoskeleton	Endoplasmic Reticulum	Endosome
Myosin-10	MYH10	P35580	0.0273	**8.0**	** C **	** Pm **		** N **	** S **		
SEC14-like protein 2	SEC14L2	O76054	0.0457	**1.8**	** C **			** N **			
Charged multivesicular body protein 5	CHMP5	Q9NZZ3	0.0081	**1.5**	** C **			** N **			** E **
14-3-3 protein gamma	YWHAG	P61981	0.0079	**1.4**	** C **	** Pm **	** M **				
Gamma-glutamylcyclotransferase	GGCT	O75223	0.0170	**1.4**	** C **						
Probable tRNA(His) guanylyltransferase	THG1L	Q9NWX6	0.0098	**1.4**	** C **		** M **				
Alcohol dehydrogenase [NADP(+)]	AKR1A1	P14550	0.0360	**1.3**	** C **	** Pm **					
14-3-3 protein beta/alpha	YWHAB	P31946-2	0.0299	**1.3**	** C **	** Pm **	** M **	** N **			
Retinal dehydrogenase 1	ALDH1A1	P00352	0.0235	**1.3**	** C **						
NHL repeat-containing protein 2	NHLRC2	Q8NBF2	0.0382	**1.2**	** C **						
Transforming protein RhoA	RHOA	P61586	0.0275	**1.2**	** C **	** Pm **			** S **	** Er **	** E **
Mannose-1-phosphate guanyltransferase alpha	GMPPA	Q96IJ6	0.0227	**−1.2**	** C **						
Gamma-enolase	ENO2	P09104	0.0494	**−1.3**	** C **	** Pm **					
Adenosine kinase	ADK	P55263	0.0230	**−1.3**	** C **	** Pm **		** N **			
Thioredoxin	TXN	P10599	0.0391	**−1.3**	** C **			** N **			
Eukaryotic translation initiation factor 2 subunit 2	EIF2S2	P20042	0.0258	**−1.5**	** C **						
Prostaglandin E synthase 3	PTGES3	Q15185-4	0.0240	**−1.5**	** C **			** N **			
Sorting nexin-6	SNX6	Q9UNH7	0.0076	**−1.5**	** C **			** N **			** E **
DnaJ homolog subfamily B member 2	DNAJB2	P25686	0.0145	**−1.6**	** C **			** N **		** Er **	
Semaphorin-7A	SEMA7A	O75326	0.0047	**−1.6**		** Pm **					
Guanine nucleotide-binding protein subunit alpha-13	GNA13	Q14344	0.0486	**−1.7**	** C **	** Pm **		** N **			
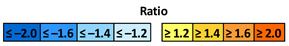

**Table 3 cells-11-00049-t003:** More and less abundant proteins in circulating reticulocytes of PV JAK2^V617F^ patients compared with healthy individuals. The density gradient was used to enrich reticulocyte count from fresh human blood samples following mature red blood and reticulocyte sorting. Whole cell proteome was analyzed using nanoscale liquid chromatography coupled to tandem mass spectrometry (MS/MS). Significantly more or less abundant proteins between groups were analyzed using Student’s *t*-test *p* < 0.05 with a protein level fold change cutoff of >1.2; *n* = 4. Polycythemia Vera (PV), healthy individuals (CT), and V617F mutation in JAK2 (JAK2). Protein cellular localization was defined using UniProt annotation and GO–cellular component data. Subcellular locations: cytoplasm (C), plasma membrane (PM), mitochondria (M), nucleus (N), cytoskeleton (S), endoplasmic reticulum (ER), ribosomes (R), Golgi apparatus (G), and endosome (E).

ReticulocytesPV JAK2 vs. CT	GeneName	AccessionNumber	*p*-Value	Fold Change	Subcellular Location
Protein	Cytoplasm	Plasma Membrane	Mitochondria	Nucleus	Cytoskeleton	Endoplasmic Reticulum	Ribosomes	Golgi Apparatus	Endosome
Transferrin receptor protein 1	TFRC	P02786	0.0439	**6.4**		** Pm **							** E **
ATP synthase subunit alpha, mitochondrial	ATP5A1	P25705	0.0012	**4.3**			** M **						
60 kDa heat shock protein, mitochondrial	HSPD1	P10809	0.0212	**3.9**			** M **						
Methionine–tRNA ligase, cytoplasmic	MARS	P56192	0.0100	**3.8**	** C **								
Prohibitin	PHB	P35232	0.0253	**3.3**	** C **	** Pm **	** M **	** N **					
Myosin-9	MYH9	P35579	0.0221	**3.1**	** C **			** N **	** S **				
Stress-70 protein, mitochondrial	HSPA9	P38646	0.0151	**2.7**			** M **						
Isocitrate dehydrogenase [NADP], mitochondrial	IDH2	P48735	0.0493	**2.4**			** M **						
Protein-L-isoaspartate(D-aspartate) O-methyltransferase	PCMT1	P22061	0.0466	**2.4**	** C **								
Vacuolar protein sorting-associated protein 35	VPS35	Q96QK1	0.0356	**2.3**	** C **								** E **
60S ribosomal protein L22	RPL22	P35268	0.0056	**2.0**	** C **			** N **			** R **		
Synembryn-A	RIC8A	Q9NPQ8-4	0.0292	**2.0**	** C **	** Pm **							
Voltage-dependent anion-selective channel protein 3	VDAC3	Q9Y277	0.0127	**2.0**			** M **	** N **					
RNA 3-terminal phosphate cyclase	RTCA	O00442	0.0355	**2.0**				** N **					
Protein disulfide-isomerase	P4HB	P07237	0.0232	**1.9**	** C **	** Pm **			** S **	** Er **		** G **	
40S ribosomal protein S7	RPS7	P62081	0.0080	**1.8**	** C **			** N **	** S **		** R **		
60S ribosomal protein L23a	RPL23A	P62750	0.0106	**1.8**	** C **			** N **			** R **		
Glucose 1,6-bisphosphate synthase	PGM2L1	Q6PCE3	0.0358	**1.8**	** C **								
Trifunctional enzyme subunit beta, mitochondrial	HADHB	P55084-2	0.0004	**1.8**			** M **						
Glutamine synthetase	GLUL	P15104	0.0127	**1.8**	** C **	** Pm **	** M **	** N **		** Er **			
40S ribosomal protein S18	RPS18	P62269	0.0494	**1.8**	** C **			** N **			** R **		
60S ribosomal protein L11	RPL11	P62913	0.0410	**1.7**	** C **			** N **			** R **		
Schlafen family member 14	SLFN14	P0C7P3-2	0.0323	**1.7**	** C **			** N **					
60S ribosomal protein L8	RPL8	P62917	0.0117	**1.7**	** C **						** R **		
60S ribosomal protein L13a	RPL13A	P40429	0.0423	**1.6**	** C **			** N **			** R **		
60S ribosomal protein L7a	RPL7A	P62424	0.0085	**1.6**	** C **			** N **			** R **		
Tyrosine-protein phosphatase non-receptor type 11	PTPN11	Q06124	0.0421	**1.6**	** C **	** Pm **	** M **	** N **					
Trifunctional purine biosynthetic protein adenosine-3	GART	P22102	0.0009	**1.6**	** C **								
Coatomer subunit delta	ARCN1	P48444	0.0006	**1.5**	** C **					** Er **		** G **	
60S ribosomal protein L12	RPL12	P30050	0.0366	**1.5**	** C **			** N **			** R **		
Argininosuccinate lyase	ASL	P04424	0.0373	**1.5**	** C **								
40S ribosomal protein S2	RPS2	P15880	0.0059	**1.5**	** C **			** N **			** R **		
Ras GTPase-activating protein-binding protein 1	G3BP1	Q13283	0.0354	**1.5**	** C **			** N **			** R **		
Aspartate–tRNA ligase, cytoplasmic	DARS	P14868	0.0360	**1.4**	** C **								
Heterogeneous nuclear ribonucleoprotein Q	SYNCRIP	O60506-3	0.0391	**1.4**	** C **			** N **		** Er **	** R **		
14-3-3 protein beta/alpha	YWHAB	P31946-2	0.0410	**1.3**	** C **	** Pm **	** M **	** N **					
Eukaryotic translation initiation factor 3 subunit J	EIF3J	O75822	0.0356	**1.3**	** C **								
Cysteine protease ATG4A	ATG4A	Q8WYN0	0.0101	**1.3**	** C **								
Double-stranded RNA-binding protein Staufen homolog 1	STAU1	O95793-2	0.0248	**1.2**	** C **	** Pm **			** S **	** Er **			
Spermine synthase	SMS	P52788	0.0244	**−1.5**	** C **								
Dipeptidyl peptidase 9	DPP9	Q86TI2-4	0.0112	**−1.5**	** C **			** N **	** S **				
Alpha-actinin-4	ACTN4	O43707	0.0044	**−1.5**	** C **			** N **	** S **		** R **		
Ribosyldihydronicotinamide dehydrogenase [quinone]	NQO2	P16083	0.0390	**−1.6**	** C **			** N **					
Anamorsin	CIAPIN1	Q6FI81-3	0.0218	**−1.7**	** C **		** M **	** N **					
Xaa-Pro dipeptidase	PEPD	P12955	0.0258	**−1.7**									
Probable E3 ubiquitin-protein ligase HECTD4	HECTD4	Q9Y4D8-5	0.0242	**−1.7**		** Pm **							
Importin-4	IPO4	Q8TEX9	0.0176	**−1.9**	** C **			** N **					
Glutathione S-transferase Mu 3	GSTM3	P21266	0.0182	**−1.9**	** C **			** N **					
Rab GDP dissociation inhibitor alpha	GDI1	P31150	0.0250	**−2.0**	** C **							** G **	
Eukaryotic initiation factor 4A-III	EIF4A3	P38919	0.0295	**−2.1**	** C **			** N **					
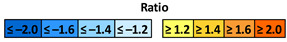

## Data Availability

The mass spectrometry proteomics data have been deposited to the ProteomeXchange Consortium via the PRIDE [63] partner repository with the dataset identifier PXD029965. The data are available via ProteomeXchange with identifier PXD029965.

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
