# Peer review of "Altered Ca2+ Homeostasis in Red Blood Cells of Polycythemia Vera Patients Following Disturbed Organelle Sorting during Terminal Erythropoiesis"

_cells, 2021, doi:10.3390/cells11010049_

Round 1
Reviewer 1 Report
The authors of the manuscript “Altered Ca2+ homeostasis in red blood cells of polycythemia vera patients following disturbed organelle sorting during terminal erythropoiesis” i) analyzed RBC membrane proteome and showed elevated ER residing proteins in polycythemia vera (PV) RBC membranes compared to control; ii) demonstrated altered calcium homeostasis in PV RBCs and cell lines with JAK2V617F mutations; iii) showed some evidence supporting that JAK2V617F may play a role in organelle retention during enucleation.
In general, the manuscript is of high interest to the field and provides important databases that may be of use in the future. The manuscript is very well written and logical. Despite the general positive outlook of this manuscript, there are several issues that need to be addressed.
1.in the results section 3.2, the authors show that RBCs from mice with JAK2 mutation were reduced in size compared to WT RBCs. It is unclear what the genetic background of the JAK2 mutated mice and WT mice are. Were control mice and JAK2 mutant mice littermates? This is important as mice with different genetic background may be associated with variations in RBC traits. Additionally, why was the MCV not used as a surrogate maker of RBC volume here? Iron deficiency could be seen in PV and can cause microcytosis; therefore, iron studies can help understand if the microcytosis is due to iron deficiency or not. Though the last point is not critical to address experimentally, at least it should be mentioned in the discussion.
2.In the spleen, there was a different in cell size all 3 states of terminal maturation, but not in the BM: What is the relevance of this?
3.JAK2 V617F was found to be associated with higher organelle remnants in mouse reticulocytes, including mitochondria. Interestingly, removal of mitochondria during erythroid maturation is mediated mostly by autophagy. Therefore, it is hard to link defected mitochondrial removal with organelle sorting defect during erythropoiesis.
3.in most experiments, changes are subtle (5-20%). Therefore, it is critical to know if the control mouse samples are isolated from littermate controls or not. Similarly, large MPN database have been published: checking MCV in JAK2 mutant versus control population would help define if JAK2 mutant MPN patient RBCs truly have altered RBS size.
4.other strategies looking at mitochondrial mass, other than mitofluor could be used to corroborate the mitofluor data (or at least discussed) to exclude the possibility of an artifact of the mitofluor.
5.some proteins were found to be more abundant in patient than control RBCs, while others were less abundant. The authors conclude that this is due to a defect in organelle sorting because many of these proteins are members of plasma membrane, golgi, endosomes, cytoskeleton, etc… however, such a conclusion is not supported. Therefore, the authors certainly need to soften their conclusion, especially that most of these proteins are localized to the cytoskeleton or nucleus, which to this reviewer truly means nothing in terms of suggesting organelle sorting defects – there is absolutely no evidence for that.
Minor comments:
-in the material and methods section, there is a typo, the JAK2 allele is floxed (instead of flexed).
Author Response
Reviewer 1
The authors of the manuscript “Altered Ca2+ homeostasis in red blood cells of polycythemia vera patients following disturbed organelle sorting during terminal erythropoiesis” i) analyzed RBC membrane proteome and showed elevated ER residing proteins in polycythemia vera (PV) RBC membranes compared to control; ii) demonstrated altered calcium homeostasis in PV RBCs and cell lines with JAK2V617F mutations; iii) showed some evidence supporting that JAK2V617F may play a role in organelle retention during enucleation.
In general, the manuscript is of high interest to the field and provides important databases that may be of use in the future. The manuscript is very well written and logical. Despite the general positive outlook of this manuscript, there are several issues that need to be addressed.
Authors: We thank the reviewer for the positive comments and important questions, which are addressed below.
1.in the results section 3.2, the authors show that RBCs from mice with JAK2 mutation were reduced in size compared to WT RBCs. It is unclear what the genetic background of the JAK2 mutated mice and WT mice are. Were control mice and JAK2 mutant mice littermates? This is important as mice with different genetic background may be associated with variations in RBC traits. Additionally, why was the MCV not used as a surrogate maker of RBC volume here? Iron deficiency could be seen in PV and can cause microcytosis; therefore, iron studies can help understand if the microcytosis is due to iron deficiency or not. Though the last point is not critical to address experimentally, at least it should be mentioned in the discussion.
Authors: We thank the reviewer for the important comment regarding the genetic background of the mice used in this study. The control mice and JAK2 mutant mice were indeed littermates. We have added this information in the manuscript (line 102).
Furthermore, the reviewer is right about the MCV as a surrogate marker of RBC volume. This has been already reported for the mice used in this study, we have added this information in the results 3.2 paragraph (lines 300 and 301). We have chosen to use the imaging approach in order to distinguish between the reticulocytes and the mature RBC populations and also to address this parameter on a single cell level both in circulating cells and differentiating erythroblasts.
In the discussion section, we mention that microcytosis in PV JAK2V617F patients are long associated with insufficient iron stores to compensate the increase of haematocrit. Importantly, we stress that haemoglobin and haematocrit values for the ET JAK2V617F patients are in the levels of healthy individuals. Our study reports microcytosis in both PV and ET JAK2V617F cells suggesting insufficient iron levels are likely not the only explanation of JAK2V617F microcytosis. Our data shows for the first time that Gárdos (KCa3.1) activity is increased in JAK2V617F context that leads to cell dehydration, making KCa3.1 an interesting target for future microcytosis investigations. Undoubtedly, further iron studies are also required to determine the role of iron in JAK2V617F cell microcytosis.
2.In the spleen, there was a different in cell size all 3 states of terminal maturation, but not in the BM: What is the relevance of this?
Authors: The reviewer is totally right about this observation. Normal steady state erythropoiesis in mice takes place in bone marrow and spleen. However, stress erythropoiesis in mice takes place predominantly in spleen. Since JAK2V617F is a cause of stress erythropoiesis, we assume that differences in the cell size and organelle remnant levels in spleen cells compared with the bone marrow are due to the predominant occurrence of stress erythropoiesis in this organ.
3.JAK2 V617F was found to be associated with higher organelle remnants in mouse reticulocytes, including mitochondria. Interestingly, removal of mitochondria during erythroid maturation is mediated mostly by autophagy. Therefore, it is hard to link defected mitochondrial removal with organelle sorting defect during erythropoiesis.
We agree with the reviewer about autophagy being the main process at the origin of mitochondria removal during erythroid maturation. In our study we report organelle retention in reticulocytes in the JAK2V617F context at the very first step following enucleation. We observe high levels of mitochondria remnants post enucleation (predominantly in the spleen) and increased levels of mitochondria in circulating reticulocytes in JAK2V617F mice. Some mitochondria are removed my mitophagy and autophagy processes during each step of reticulocyte maturation (Figure 5). However, we think that the presence of increased levels of mitochondria remnants in the final reticulocyte maturation stage (circulating reticulocytes) suggests defects in mitochondrial removal or that mitochondria removal machinery is overwhelmed.
3.in most experiments, changes are subtle (5-20%). Therefore, it is critical to know if the control mouse samples are isolated from littermate controls or not. Similarly, large MPN database have been published: checking MCV in JAK2 mutant versus control population would help define if JAK2 mutant MPN patient RBCs truly have altered RBS size.
Authors: Control mice and JAK2 mutant mice were littermates, this is now stated in the manuscript (line 102). We agree with the reviewer about the MCV being a surrogate marker of RBC microcytosis in MPN. Microcytosis in PV is well established, it is discussed in our answer to the first question of the reviewer.
4.other strategies looking at mitochondrial mass, other than mitofluor could be used to corroborate the mitofluor data (or at least discussed) to exclude the possibility of an artifact of the mitofluor.
Authors: We agree with the reviewer about this important comment. Multiple mitochondria markers could be used to study their amounts in the cells. However, the aim of our study was to use organelle markers to study organelle retention and removal post enucleation instead of investigating mitochondria specifically. We chose mitofluor (mitochondria marker) and thiazole orange (ribosomal RNA marker). In our study we do use a second and more global strategy to address the organelle levels in the RBCs and to exclude the possibility of artifacts related to the fluorescent dyes, i.e. a proteomic approach. Our proteomic study confirms elevated levels of organelle remnants in PV JAK2V617F circulating reticulocytes, including mitochondrial and ribosomal proteins, supporting the mitofluor and thiazole orange data. It would be, however, curious to further investigate mitochondria sorting, mitophagy and mitochondria activity for which abundance of various mitochondria markers should be used.
5.some proteins were found to be more abundant in patient than control RBCs, while others were less abundant. The authors conclude that this is due to a defect in organelle sorting because many of these proteins are members of plasma membrane, golgi, endosomes, cytoskeleton, etc… however, such a conclusion is not supported. Therefore, the authors certainly need to soften their conclusion, especially that most of these proteins are localized to the cytoskeleton or nucleus, which to this reviewer truly means nothing in terms of suggesting organelle sorting defects – there is absolutely no evidence for that.
Authors: We agree with the reviewer that the conclusion was overdrawn. We have addressed this point by stating that the proteomics data suggests alterations during erythroid differentiation in a JAK2V617F context (line 442).
Minor comments:
-in the material and methods section, there is a typo, the JAK2 allele is floxed (instead of flexed).
Authors: We thank the reviewer for careful reading, the typo is corrected in the final version of the manuscript.
Reviewer 2 Report
In this paper, Buks and colleagues present data to show alterations in calcium homeostasis and RBC development in cases of JAK2 V617F mutations. The authors show increased intracellular calcium in PV patients and JAK2 V617F mice, as well as problems during the maturation of RBCs related to organelle sorting and enucleation. Given the frequency of this mutation in PV patients, this is an interesting and important study.
The paper is well written, the methods are extensive and comprehensive. The conclusions are supported by the data, and for most conclusions, they are supported by multiple experimental approaches. I only have some minor points to make:
- The authors use Forward Scatter as a measure of cell surface area. Did the authors perform any analysis on the results to exclude doublets?
- Did the authors notice any changes in SSC on flow cytometry? One would assume that in cells with 'retained' organelles, there would also be an increase in Side Scatter. I don't ask this from a critical point of view (there is plenty enough data to show altered organelle sorting) but it would be interesting to see if this could be a quicker measure to identify this phenomena. If the authors do observe it, it may be worth mentioning.
- While the authors specify they have performed Mann-Whitney or Wilcoxon tests, some graphs clearly make multiple comparisons and there is no description of the method used to compensate for these comparisons. The authors should specify which correction has been applied (if one has).
- In Figure 1, the authors refer to the cells as Fluo4 positive based on the position of the gate on CT cells, but presumably the cells in the peak on the CT are actually positive for Flou4 (as in they have free intracellular calcium). I think the MFI in this instance (which also shows an increase) makes more sense. I would consider changing the terminology used to describe these Fluo4"+" cells when presenting them as a percentage. This is a minor point; there is sufficient other data provided that makes it clear that intracellular Ca2+ is raised in PV RBCs.
Author Response
Reviewer 2
Authors: We thank the reviewer for their positive comments and important questions, which are addressed bellow.
In this paper, Buks and colleagues present data to show alterations in calcium homeostasis and RBC development in cases of JAK2 V617F mutations. The authors show increased intracellular calcium in PV patients and JAK2 V617F mice, as well as problems during the maturation of RBCs related to organelle sorting and enucleation. Given the frequency of this mutation in PV patients, this is an interesting and important study.
The paper is well written, the methods are extensive and comprehensive. The conclusions are supported by the data, and for most conclusions, they are supported by multiple experimental approaches. I only have some minor points to make:
- The authors use Forward Scatter as a measure of cell surface area. Did the authors perform any analysis on the results to exclude doublets?
Authors: We confirm that doublets were excluded from the flow cytometry analysis.
- Did the authors notice any changes in SSC on flow cytometry? One would assume that in cells with 'retained' organelles, there would also be an increase in Side Scatter. I don't ask this from a critical point of view (there is plenty enough data to show altered organelle sorting) but it would be interesting to see if this could be a quicker measure to identify this phenomena. If the authors do observe it, it may be worth mentioning.
Authors: This is a very interesting suggestion. We did not observe significant changes in SSC in JAK2V617F context likely because of abundance of haemoglobin and not detectable increase of retained organelles in reticulocytes using this method.
- While the authors specify they have performed Mann-Whitney or Wilcoxon tests, some graphs clearly make multiple comparisons and there is no description of the method used to compensate for these comparisons. The authors should specify which correction has been applied (if one has).
Authors: We thank the reviewer for this comment, we have made the comparisons clearer in the figure legends.
- In Figure 1, the authors refer to the cells as Fluo4 positive based on the position of the gate on CT cells, but presumably the cells in the peak on the CT are actually positive for Flou4 (as in they have free intracellular calcium). I think the MFI in this instance (which also shows an increase) makes more sense. I would consider changing the terminology used to describe these Fluo4"+" cells when presenting them as a percentage. This is a minor point; there is sufficient other data provided that makes it clear that intracellular Ca2+ is raised in PV RBCs.
Authors: It is true that all cells have some free Ca2+ levels, but very low levels (present in the vast majority of the RBCs) are below the detection limit of Fluo4. For this reason, the peak of the cells is overlaying with the background. Instead, we chose to compare the ‘true’ positive population, and continue our study using more specialized methods measuring free calcium levels (e.g. studying the activity of Gardos channel).